# Estimating waiting times, patient flow, and waiting room occupancy density as part of tuberculosis infection prevention and control research in South African primary health care clinics

**Aaron S. Karat**[1,2]*, **Nicky McCreesh**[1], **Kathy Baisley**[1,3], **Indira Govender**[1], **Idriss I. Kallon**[4,5], **Karina Kielmann**[2], **Hayley MacGregor**[6], **Anna Vassall**[1], **Tom A. Yates**[7], **Alison D. Grant**[1,3,8]

1 TB Centre, London School of Hygiene & Tropical Medicine, London, United Kingdom, 2 The Institute for Global Health and Development, Queen Margaret University, Edinburgh, United Kingdom, 3 Africa Health Research Institute, School of Laboratory Medicine & Medical Sciences, College of Health Sciences, University of KwaZulu-Natal, Durban, South Africa, 4 Division of Social and Behavioural Sciences, Faculty of Health Sciences, University of Cape Town, Cape Town, South Africa, 5 Centre for Evidence-Based Health Care, Division of Epidemiology and Biostatistics, Department of Global Health, Faculty of Medicine and Health Sciences, Stellenbosch University, Cape Town, South Africa, 6 The Institute of Development Studies, University of Sussex, Brighton, United Kingdom, 7 Department of Infectious Disease, Faculty of Medicine, Imperial College London, London, United Kingdom, 8 School of Public Health, Faculty of Health Sciences, University of the Witwatersrand, Johannesburg, South Africa

* aaron.s.karat@gmail.com

## Abstract

Transmission of respiratory pathogens, such as *Mycobacterium tuberculosis* and severe acute respiratory syndrome coronavirus 2, is more likely during close, prolonged contact and when sharing a poorly ventilated space. Reducing overcrowding of health facilities is a recognised infection prevention and control (IPC) strategy; reliable estimates of waiting times and 'patient flow' would help guide implementation. As part of the *Umoya omuhle* study, we aimed to estimate clinic visit duration, time spent indoors versus outdoors, and occupancy density of waiting rooms in clinics in KwaZulu-Natal (KZN) and Western Cape (WC), South Africa. We used unique barcodes to track attendees' movements in 11 clinics, multiple imputation to estimate missing arrival and departure times, and mixed-effects linear regression to examine associations with visit duration. 2,903 attendees were included. Median visit duration was 2 hours 36 minutes (interquartile range [IQR] 01:36–3:43). Longer mean visit times were associated with being female (13.5 minutes longer than males; p<0.001) and attending with a baby (18.8 minutes longer than those without; p<0.01), and shorter mean times with later arrival (14.9 minutes shorter per hour after 0700; p<0.001). Overall, attendees spent more of their time indoors (median 95.6% [IQR 46–100]) than outdoors (2.5% [IQR 0–35]). Attendees at clinics with outdoor waiting areas spent a greater proportion (median 13.7% [IQR 1–75]) of their time outdoors. In two clinics in KZN (no appointment system), occupancy densities of ~2.0 persons/m² were observed in smaller waiting rooms during busy periods. In one clinic in WC (appointment system, larger waiting

**Data Availability Statement:** Data are available from the ReShare repository (http://doi.org/10.5255/UKDA-SN-854435).

**Funding:** The support of the Economic and Social Research Council (UK) is gratefully acknowledged. The Umoya omuhle study is funded by the Antimicrobial Resistance Cross Council Initiative supported by the seven research councils in partnership with other funders including support from the GCRF (ref. ES/P008011/1). Additional support was received from The Bloomsbury SET (Research England; ref. CCF17-7779). TAY is funded via an NIHR Academic Clinical Fellowship and acknowledges support from the NIHR Imperial Biomedical Research Centre. The funders had no role in study design, data collection and analysis, decision to publish, or preparation of the manuscript.

**Competing interests:** The authors have declared that no competing interests exist.

**Abbreviations:** ART, antiretroviral therapy; CI, confidence interval; hh, hours; IPC, infection prevention and control; IQR, interquartile range; KZN, Kwa-Zulu Natal; mm, minutes; *Mtb*, *Mycobacterium tuberculosis*; NCD, noncommunicable disease; PHC, primary health care; REDCap, Research Electronic Data Capture; ref, reference; SARS-COV-2, severe acute respiratory syndrome coronavirus 2; TB, tuberculosis; WC, Western Cape.

areas), occupancy density did not exceed 1.0 persons/m$^2$ despite higher overall attendance. In this study, longer waiting times were associated with early arrival, being female, and attending with a young child. Occupancy of waiting rooms varied substantially between rooms and over the clinic day. Light-touch estimation of occupancy density may help guide interventions to improve patient flow.

# Background

Transmission of respiratory infections is a persistent problem in health care facilities: proportions of attendees who are infectious and susceptible are likely to be higher than in other settings [1–3]. As well as creating risk for individuals attending for care, nosocomial transmission can 'institutionally amplify' epidemics and represents a serious threat to health worker safety [4–7]. Pathogens such as *Mycobacterium tuberculosis* (*Mtb*) [8, 9] and severe acute respiratory syndrome coronavirus 2 (SARS-CoV-2) [10, 11] are more likely to be transmitted during close, prolonged contact and between individuals 'sharing air' in a poorly ventilated space [12, 13].

Reducing overcrowding and time spent in health care facilities are recommended infection prevention and control (IPC) measures for tuberculosis (TB) and other respiratory infections [14, 15]. Several initiatives to reduce frequency of clinic visits have been tested and deployed, primarily with the intention of providing 'differentiated care' [16, 17]. Interventions that focus on the movement of people around the facility ('patient flow'), however, have received less attention than individual-focused IPC interventions, such as mask-wearing, triage, and prompt initiation of treatment [18, 19], and 'passive' interventions, such as structural changes to improve ventilation [20]. This is partly because of the complexity of intervening to change patient flow in busy facilities, which can vary widely in size, layout, internal organisation, and patient load [21], and because it is difficult to measure changes in flow. Considerations also differ for hospitals and primary health care (PHC) clinics; this article relates mainly to operations at PHC level.

## Estimating time spent in South African PHC clinics

PHC clinics in South Africa are the main interface between the public health system and the population. Most services at PHC level are nurse-led and include the majority of routine care for pregnant women, children, and people living with HIV, drug-sensitive TB, and non-communicable diseases. PHC clinics also provide a range of acute services (from minor injuries to infections) and voluntary testing for HIV and TB. Some larger facilities (offering a 24-hour maternity service, casualty, and a short stay ward) are known as community health centres (CHCs), though these services are also offered by some PHC clinics. In this article, the term 'PHC clinic' is used to include CHCs unless otherwise specified.

The nearly 3,500 PHC clinics in South Africa function in diverse epidemiological, political, cultural, and climactic conditions and serve individuals with a wide range of needs [22]. Long clinic waiting times have been documented over many years [23, 24] and are frequently cited as a concern for patients [25–27]. The 'Ideal Clinic' initiative, devised and scaled up by the National Department of Health since 2013, aims to enable universal standards of practice, routine measurement of relevant metrics, and fair comparisons of performance between facilities [28]. Regular estimation of waiting times is recommended by Ideal Clinic and is standard practice in most clinics, though the methods used vary by province. South African national guidelines recommend that clinic visits should take less than three hours [29].

Most approaches to estimating waiting times conceptualise the patient's journey through the clinic as linear, with each individual passing through the clinic as quickly as possible while ensuring that the necessary service points are accessed. In South Africa, waiting times are usually measured through the provision of a physical card to all or a selection of patients attending on the day [30]. This card is time-stamped at the beginning and end of each interaction with a service point (for example, when an individual has their blood pressure measured or sees a clinician). This method is useful for estimating time spent waiting for services and the efficiency of selected processes but is less useful in assessing risk of respiratory disease transmission, as it does not describe where patients are waiting. This method also does not include non-patient attendees (e.g., parents accompanying children), any of whom may be susceptible to infection or have undiagnosed disease, nor does it allow for estimation of staff exposure to 'high risk' areas.

This work was conducted as part of the *Umoya omuhle* study, a multidisciplinary initiative taking a 'whole systems' approach to TB IPC in South African PHC clinics [31]. This study component aimed to develop and test a method to 1) estimate how long attendees spent in clinics, and determine why some individuals spent longer than others; 2) estimate how long attendees spend in outdoor (lower risk) and indoor (higher risk) clinic areas; 3) describe variation in occupancy of waiting areas over the clinic day; and 4) collect data for mathematical modelling of IPC interventions in clinics to reduce risk of *Mtb* transmission [32].

## Methods

The literature was reviewed to explore methods previously used to estimate waiting times, occupancy density, crowding, and patient flow (Appendix 1 in S1 Text). The methods that allowed estimation of the broadest range of outcomes were waiting/working time surveys (either paper-based or using a real-time location system) [23, 27, 33] and camera-based systems [34–36]. Given the resources available to the study, as well as the ethical and logistical complications of using camera-based systems in multiple public clinics, an approach based on waiting time surveys was used. As described below, unique barcodes and hand-held barcode scanners were used instead of radio-frequency identification tags [37–39] or a paper-based system [30].

### Data collection

Data were collected in clinics in KwaZulu-Natal (KZN) province (coded KZN1–6) from 22 February to 14 March 2019 and in Western Cape (WC) province (coded WC1–6) from 14–22 May 2019. Clinics were visited at least once prior to data collection to provide written information, discuss procedures with the manager and staff, and observe patient flow (Fig A in S1 Text). On the day of data collection, a member of the research team attended the morning staff meeting to answer additional questions and issue unique barcodes to staff (including non-clinical staff). Each health worker who accepted a barcode was asked for their job title and role that day. No other personal information was collected. Perforated cards were used that could easily be divided into two after completion to protect confidentiality (Fig B in S1 Text). At the top of the card was a unique one-dimensional (1D) barcode and brief instructions. The rest of the card contained a brief questionnaire and the same 1D barcode. After questionnaire completion, the card was divided: the bottom part was returned to the researcher and the top part retained by the participant (or health worker) to be worn, on a lanyard, around their neck.

Researchers (usually 6–10 individuals, depending on the size of the clinic) were positioned at key points throughout the premises, including facility entrance(s), the window where patient files were issued (filing station), the room or space where blood pressure and other

measurements were taken (triage/vitals station), and doorways of consultation rooms and waiting areas (Fig C in S1 Text). Each researcher carried a cordless, hand-held barcode scanner (OPN-2001, Opticon Limited, United Kingdom); clinic staff in certain locations (most often consultation rooms) were also asked to carry scanners and were instructed on how and when to use them. Scanners were cleaned of all data and time-synchronised before each day's data collection.

All individuals attending the clinic during the hours of data collection were asked to participate. A researcher approached attendees and explained the purpose of the study and that participation was voluntary and anonymous. Numbers and details of individuals who refused were not recorded because the enrolment process was time-sensitive. Individuals who agreed to participate were asked to complete the card (recording their sex, age group, and reason for attendance) and to wear the barcode until they left the clinic. If requested, the researcher completed the card on a participant's behalf. Individuals attending together had their cards stapled together before storage; this was accounted for during data entry.

At the beginning of data collection, all individuals already in the clinic were asked to participate and their location noted by use of a designated scanner. Simultaneously, researchers positioned at the entrance(s) asked individuals who were entering the clinic to participate. Within 60–90 minutes of commencing data collection, all individuals were offered the opportunity to participate; this sometimes took longer for busier clinics.

Barcode scanners at doorways and other designated 'transition points' were used to scan every person with a barcode who passed through. Scanners at filing stations, vitals stations, consultation rooms, or at other service points were used to scan an individual's arrival and departure from that station or room. Individuals leaving the clinic had their barcode scanned as they left. For logistical reasons, data collection was stopped at 1400; at this point, the barcodes of all individuals remaining in the clinic were scanned, with designated scanners used to note their location. A questionnaire was administered to the facility manager or nurse in charge to record information about staffing levels and other factors that may have affected service provision. The dimensions of waiting areas were measured using a Bosch PLR 40R digital laser measure (Bosch, Gerlingen, Germany; accuracy +/-2 mm).

### Data management

Data from barcode scanners were transferred to a password-protected computer at the end of each day's data collection. Data from completed cards were entered into a Research Electronic Data Capture (REDCap) database [40, 41], hosted at the Africa Health Research Institute, that was programmed to assign a 'group ID' to all individuals attending together (denoted by cards having been stapled together). Data from questionnaires administered to facility managers were entered into a password-protected spreadsheet.

### Analysis

Analysis had three strands, examining 1) time spent in clinic and factors that influenced this; 2) the proportion of each individual's time in clinic that was spent in indoor spaces (higher transmission risk) vs. outdoors (lower risk, primarily due to better ventilation), and how this varied by clinic and reason for visit; and 3) in clinics with more than one indoor waiting area, occupancy density of each indoor waiting area and how this varied over the course of the day. Table C in S1 Text describes the clinics and numbers of individuals included in each analysis.

**Time spent in clinic.** A large number of data were missing for arrival and departure times because some individuals were either 1) already in the clinic when data collection began or 2) still present in the clinic when data collection ended. These individuals therefore did not have

their arrival and/or departure time recorded. Multiple imputation was used to generate arrival and/or departure times for individuals in whom one or both was not recorded (see Appendix 2.2.1 in S1 Text for details).

Data were excluded from this analysis from clinics where clinic entry and exit were not recorded consistently (because the research team was too small to monitor all entrances and exits of the clinic). Multiply-imputed data (n = 20 imputations) were used for this analysis. Relationships between individual characteristics and time spent in clinic (continuous outcome) were examined using a mixed-effects linear regression model with a random effect for clinic. Province was included as fixed effect. The shape of the relationship between time of arrival and time spent in clinic was examined using fractional polynomials regression with a set of defined powers ($-2$, $-1$, $-0.5$, $0.5$, $1$, $2$, and $\ln[x]$) and a maximum of two power terms in the model. The differences in model deviances were compared: the linear model was used if the improvement in fit was not statistically significant at $p < 0.05$. Province, age group, sex, and the ratio of patients to clinical staff were included in the multivariable model as a priori confounders; other variables were included if they showed an important association ($p < 0.05$) in the univariable model. Coefficients, representing the difference in mean time spent in clinic (in minutes), are reported with 95% confidence intervals (CIs).

**Proportion of time spent indoors vs. outdoors.** Non-imputed data were used. Data were excluded 1) from clinics where a scanner had not been positioned at all facility entrances and all indoor/outdoor doorways and 2) from individuals with a total captured visit time of less than five minutes and no record of having left the clinic, as they were considered likely to have discarded their barcode (Table C in S1 Text). Each individual's pathway through the clinic was mapped: for each barcode scan recorded, the individual's location in the time preceding the scan was categorised as 'indoors', 'outdoors', or 'unknown' (if they appeared to have moved between two unconnected locations, indicating a missing barcode scan) based on the location of their previous scan. Total time spent in each type of location (as a proportion of the individual's overall recorded time in clinic) was summarised by clinic and by self-reported reason for clinic attendance.

**Occupancy density.** Non-imputed data were used from clinics that had more than one indoor waiting area and where a barcode scanner had been positioned at all entrances and exits of at least two waiting areas. Data were divided into 10 second slices and entries and exits from each demarcated space noted for each 10 second period; the number of individuals within a space at the end of each 10 second period was divided by the floor area and volume of that space to give the occupancy density (in persons/m$^2$ and persons/m$^3$, respectively) for that 10 second period.

Analyses were conducted in Stata versions 14 and 16 (Statacorp, College Station, Tx). Figures were created using Stata, Microsoft PowerPoint, Microsoft Excel, and Inkscape v0.92.4 [42].

## Ethical considerations

Identifiable data were not collected from participating individuals; written informed consent was therefore not requested. This study received ethical approval (including waivers for written informed consent) from the Biomedical Research Ethics Committee of the University of KwaZulu-Natal (ref. BE082/18), the Human Research Ethics Committee of the Faculty of Health Sciences of the University of Cape Town (ref. 165/2018), the Research Ethics Committee of Queen Margaret University (ref. REP 0233), and the Observational/Interventions Research Ethics Committee of the London School of Hygiene & Tropical Medicine (ref. 14872).

## Results

Patient flow in study clinics was broadly organised around three key steps in the following order: 1) patient registration and file collection, 2) vital signs, and 3) health worker consultation. Individuals usually waited in a different part of the clinic for each step. Paths taken through the clinic depended on the reason for visit (many individuals also visited one or more of the in-clinic pharmacy, phlebotomist, and other specialist practitioners) and varied between clinics based on their size, design, and organisation of care. In most clinics, individuals attending for TB care (i.e., those being investigated for TB or taking anti-TB treatment) were 'fast-tracked' and bypassed steps 1 and 2 above. Clinics varied widely in size, population served, services offered, and organisation of care. Importantly, some clinics routinely asked patients to wait in covered outdoor waiting areas, whereas others had only indoor areas or did not use their outdoor areas as part of a designated patient pathway. All clinics in WC and no clinics in KZN operated a date-time appointment system for at least some patients (Table E in S1 Text); no clinics operated an active queue management system.

Twelve datasets were available for analysis from 11 clinics: six in KZN and five in WC (Table 1; clinic WC4 could not be visited for logistical reasons and clinic KZN1 was visited for a second time to attempt better coverage). Eight facilities were PHC clinics and three were CHCs (Table E in S1 Text). Data were collected for 2,903 patients and visitors: 1,925 (66.3%) in KZN and 978 (33.7%) in WC. Across clinics, a median 70% (interquartile range [IQR] 69%–74%) of clinic attendees were female. Most individual characteristics were similar between provinces, with the only large differences seen in 'main reason for clinic visit': in KZN clinics, a median 32.6% of clinic attendees reported attending for HIV care or antiretroviral therapy (ART), compared with a median 3.5% in WC clinics. This was thought likely due, at least in part, to an error during data collection in WC clinics, with 'acute care' consistently incorrectly marked by those attending for HIV care (49% in WC vs. 29% in KZN). Because no identifying details of individuals were collected, this could not be rectified, and the two categories were combined for the regression analysis (but are shown separately in Table 1 for completeness).

### Time spent in clinics

Data were excluded from clinic KZN4 (n = 269) as all entrances and exits had not been monitored. Data from 2,634 individuals attending 10 clinics (11 data collection exercises) underwent multiple imputation and were included in this analysis (1,063 [40%] missing time of arrival and 934 [35%] missing time of departure; Table D in S1 Text). Overall median time spent in clinic was 2 hours 36 minutes (IQR 01:36–3:43). This was similar in each province (KZN 02:33 [IQR 01:35–3:40; n = 1,656]; WC 02:42 [IQR 01:37–03:49; n = 978]). Visit durations by demographics and reason for visit are provided in Table F in S1 Text.

In univariable analysis (Table 2), there was strong evidence of an increase in mean time spent in clinic for individuals who were female (p <0.001), attending with a baby (p <0.001), or attending with ≥1 other person (p <0.01). There was also strong evidence of differences by reason for visit (p <0.01): individuals attending for TB care and ante/post-natal care spent the shortest time in clinic. Mean time in clinic reduced by ~15 minutes for each hour that arrival was delayed after 0700 (p <0.001).

In multivariable analysis, longer mean times remained associated with being female (13.5 [95% CI 6–21] minutes longer than males) and attending with a baby (18.8 [95% CI 8–30] minutes longer than those attending without). Reason for visit (p <0.01) and time of arrival (p <0.001) also remained important: those attending for TB care or ante/post-natal care spent a mean 24.8 (95% CI 9–41) minutes and 32.6 (95% CI 11–54) minutes less in clinic, respectively,

**Table 1. Characteristics of clinics, individuals attending, and staff working on the day of data collection, overall and by province (N = 12 exercises at 11 clinics; N = 2,903 attendees).**

| Characteristic | All clinics, n | KZN province, n (row %) | WC province, n (row %) |
|---|---|---|---|
| Number of clinics | 11 | 6 (54.5) | 5 (45.5) |
| Number of data collection exercises | 12‖ | 7 (58.3) | 5 (41.7) |
| Hours of data collection, *HH:MM* | 77:02 | 44:32 (57.8) | 32:30 (42.2) |
| Patients & visitors included | 2,903 | 1,925 (66.3) | 978 (33.7) |
| ***On the day of data collection*** | ***Median (range) per exercise*** | ***Median (range) per exercise*** | ***Median (range) per exercise*** |
| Hours of data collection, *HH:MM* | 06:15 (05:37–07:18) | 06:15 (05:40–07:08) | 06:15 (05:37–07:18) |
| Clinical staff working*, n | 16 (3–45) | 14 (8–45) | 17 (3–45) |
| Administrative staff working*, n | 11 (4–21) | 10 (7–17) | 16 (4–21) |
| Patients† per clinical staff*, n | 14 (5–27) | 14 (6–27) | 14 (5–18) |
| Patients and visitors included, n | 252 (69–417) | 269 (170–417) | 144 (69–337) |
| Proportion female, % | 70.0 (56.3–789.7) | 71.2 (68.4–74.8) | 69.2 (56.3–79.7) |
| Proportion aged | | | |
| 0–5 years, % | 9.1 (0.7–30.8) | 8.4 (7.1–10.1) | 10.4 (5.6–30.8) |
| 6–15 years, % | 3.4 (0–9.4) | 3.5 (1.5–6.2) | 3.2 (0–8.3) |
| 16–25 years, % | 17.3 (6.3–25.7) | 20.1 (15.9–25.3) | 15.0 (13.1–17.4) |
| 26–35 years, % | 27.3 (19.4–36.2) | 27.8 (25.8–32.0) | 22.5 (19.4–36.2) |
| 36–45 years, % | 18.6 (15.0–35.7) | 18.0 (15.3–24.8) | 20.3 (15.0–21.1) |
| >45 years, % | 18.0 (7.2–35.4) | 18.4 (17.1–22.1) | 16.9 (7.2–35.4) |
| Proportion attending with a baby or very young child, % | 11.5 (0.7–36.2) | 11.2 (0.7–17.9) | 12.1 (10.4–36.2) |
| Proportion attending with ≥1 other person‡, % | 24.9 (1.5–60.8) | 22.9 (1.5–35.9) | 27.9 (15.4–60.8) |
| Proportion attending for | | | |
| Acute care/minor problems, % | 37.0 (9.3–53.5) | 28.8 (9.3–41.7) | 48.7 (34.8–53.5)¶ |
| HIV care/ART, % | 16.1 (0.8–85.9) | 32.6 (14.2–85.9) | 3.5 (0.8–7.1)¶ |
| Tuberculosis, % | 3.8 (0.6–16.3) | 2.2 (0.6–16.3) | 9.0 (2.1–13.3) |
| NCDs (including mental health), % | 4.4 (0–16.9) | 4.1 (0.4–5.3) | 6.9 (0–16.9) |
| Mother & child§, % | 12.4 (0.7–30.4) | 14.7 (0.7–19.7) | 7.7 (4.2–30.4) |
| Maternal & obstetric, % | 2.9 (0–8.7) | 3.2 (0–4.7) | 2.7 (0–8.7) |
| Accompanying a patient, % | 14.6 (1.5–22.5) | 12.4 (1.5–17.1) | 19.2 (10.1–22.5) |
| Attending for another person, % | 2.6 (0–6.3) | 2.9 (0–4.3) | 1.7 (0–6.3) |

*Based on questionnaire administered to manager or senior member of staff; data from clinic KZ04 (including number of staff) captured only for HIV/chronic unit

†Counted as those who reported a main visit reason that was not 'accompanying' or 'attending for another person'

‡Not including babies and very young children

§Includes attendance for family planning

‖Two exercises conducted at clinic KZ01, roughly one month apart

¶Disparity with KZN is likely due to an error during data collection (see text). 'HIV care/ART' combined with 'Acute care/minor problems' for subsequent analysis.

ART: antiretroviral therapy; KZN: KwaZulu-Natal; NCD: noncommunicable disease; WC: Western Cape

than those attending for HIV/acute care, and mean time in clinic reduced by 14.9 (95% CI 13–17) minutes for each hour that arrival was delayed after 0700. The results of the fractional polynomial models showed that the linear model adequately described the relationship between the time at clinic and arrival time (Appendix 3.2.1 in S1 Text).

## Proportion of time spent indoors vs. outdoors

The 2,190 clinic attendees included in this analysis (≥5 minutes captured; 10 visits; 9 clinics) spent a median 95.6% (IQR 45.6–100) of their time indoors and a median 2.5% (IQR 0–35.3)

**Table 2. Results of univariable and multivariable mixed-effects linear regression using imputed data, showing effects of different factors on total time spent in clinic (n = 2,634; 11 exercises in 10 clinics).**

| Variable | n | Univariable analysis* | | Multivariable analysis* | |
| --- | --- | --- | --- | --- | --- |
| | | Difference in time spent, minutes (95% CI) | p | Difference in time spent, minutes (95% CI) | p |
| **Province** | | | | | |
| KwaZulu-Natal | 1,656 | REF | 0.998 | REF | 0.734 |
| Western Cape | 978 | −0.03 (−32.2, 32.2) | | −5.1 (−34.8, 24.5) | |
| **Sex** | | | | | |
| Male | 783 | REF | <0.001 | REF | <0.001 |
| Female | 1,851 | **17.0 (9.4, 24.5)** | | **13.5 (6.0, 21.0)** | |
| **Age group** | | | | | |
| <16 years | 381 | REF | 0.250 | REF | 0.332 |
| 16–45 years | 1,703 | −7.0 (−17.8, 3.7) | | −5.6 (−17.2, 5.9) | |
| ≥46 years | 550 | −10.3 (−22.2, 1.6) | | −9.9 (−22.7, 2.9) | |
| **Patients† to clinical staff‡ ratio** | | | | | |
| <10:1 | 698 | REF | 0.734 | REF | 0.821 |
| ≥10:1 | 1,936 | −6.2 (−41.9, 29.5) | | −3.8 (−36.9, 29.3) | |
| **Attending with a baby or child aged less than ~15 months** | | | | | |
| No | 2,271 | REF | <0.001 | REF | 0.002 |
| Yes | 344 | **25.6 (15.2, 36.0)** | | **18.8 (8.1, 29.6)** | |
| Not recorded | 19 | 17.0 (−23.1, 57.1) | | 10.0 (−28.8, 48.8) | |
| **Attending with ≥1 other person§** | | | | | |
| No | 1,983 | REF | 0.004 | REF | 0.076 |
| Yes | 651 | **12.6 (4.0, 21.2)** | | 8.9 (−0.9, 18.7) | |
| **Time of arrival** | | | | | |
| Per hour later than 07h00 | 2,634 | **−15.1 (−17.1, −13.1)** | <0.001 | **−14.9 (−16.9, −12.9)** | <0.001 |
| **Reason for visit** | | | | | |
| Acute care/HIV care | 1,526 | REF | 0.002 | REF | 0.008 |
| Tuberculosis | 145 | **−27.2 (−43.8, −10.6)** | | **−24.8 (−40.6, −8.9)** | |
| NCDs (incl. mental health) | 157 | −9.1 (−24.4, 6.2) | | −5.9 (−20.8, 9.0) | |
| Mother & child (incl. family planning) | 297 | 9.5 (−2.6, 21.6) | | −3.6 (−15.7, 8.5) | |
| Ante/post-natal | 66 | **−22.1 (−44.4, 0.1)** | | **−32.6 (−54.0, −11.2)** | |
| Accompanying | 360 | 2.9 (−7.9, 13.8) | | −7.3 (−18.7, 4.2) | |
| Attending on another's behalf | 79 | −15.0 (−36.2, 6.3) | | −11.6 (−32.0, 8.7) | |
| Not recorded | 4 | 40.5 (−48.6, 129.5) | | 29.8 (−55.4, 115.1) | |

*Mixed-effects linear regression with a random effect for clinic day (i.e., two visits to clinic 1 treated as separate clusters).

†Attendees who reported a main visit reason that was not 'accompanying' or 'attending for another person'

‡Includes enrolled ('staff') and professional nurses, clinical nurse practitioners, clinical and enrolled nursing assistants, and doctors. Does not include lay-counsellors, peer navigators, community health workers, pharmacists, or nursing/medical students.

§Not including babies and very young children

ART: antiretroviral therapy; CI: confidence interval; incl.: including; REF: reference; NCD: non-communicable diseases

outdoors (Table G in S1 Text), with the remainder in unknown locations (locations were classed as 'unknown' when it could not be assessed if the individual was indoors or outdoors–see Methods). This varied by clinic (Fig 1A): in four clinics with an outdoor waiting area that was used as part of a designated patient pathway, individuals spent a median 13.7% (IQR 1.4–74.5; n = 1,362) of their time outdoors, compared with a median 0% (IQR 0–1.4; n = 828)

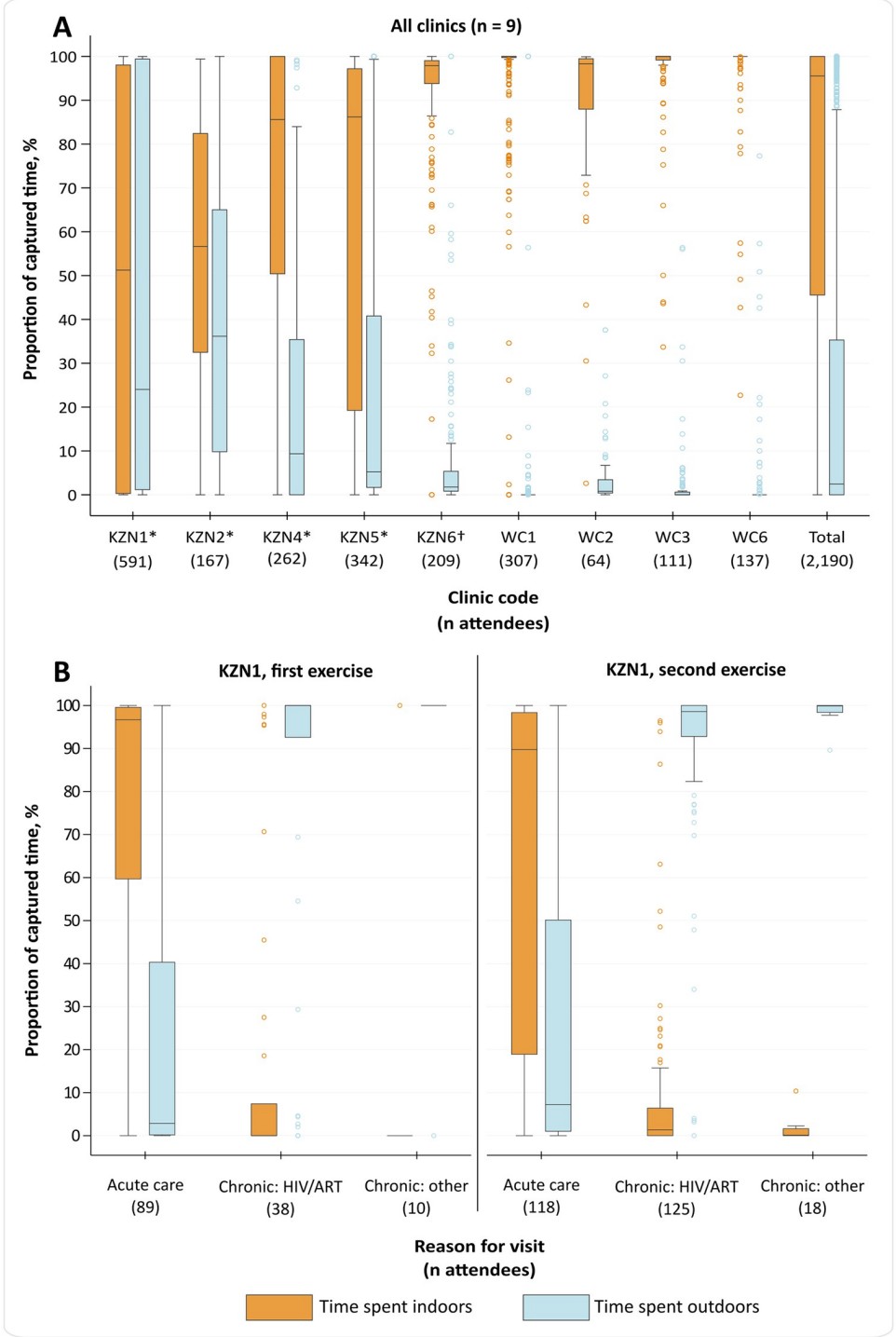

**Fig 1. Box and whiskers plots showing proportions of time spent indoors and outdoors, by clinic (panel A) and for two visits to clinic KZN1, by selected reasons for visit (panel B).** *Clinic has at least one outdoor waiting area that is part of a designated patient pathway. †Clinic has at least one outdoor waiting area, but it is not part of a designated patient pathway. The central horizontal line represents the median value; boxes represent the interquartile range (IQR); and whiskers represent largest and smallest values within 1.5 IQR of the upper and lower quartiles, respectively. Time spent in unknown locations was negligible for most clinics and is therefore not shown. **Panel A:** Proportions are shown by clinic for all attendees at nine clinics with at least five minutes captured. Data from both data collection exercises at clinic KZN1 are shown combined. **Panel B:** Proportions are shown by self-reported reason for attendance for individuals with at least five minutes captured who were attending clinic KZN1 for the three selected reasons. ART: antiretroviral therapy; KZN: KwaZulu-Natal; TB: tuberculosis; WC: Western Cape.

outdoors among attendees at the five clinics that did not have an outdoor waiting area or where the outdoor area was not used.

In clinics with outdoor waiting areas, the wide IQR (1.4–74.5) for estimated time spent outdoors reflects the considerable variation seen among attendees to clinic KZN1, where the outdoor waiting area was used only by individuals in the 'chronic' stream (Fig 1B). For example, in the second exercise at clinic KZN1, individuals attending for 'acute' care spent a median 89.8% (IQR 18.9–98.3; n = 118) of their time *indoors*, compared with those attending for HIV care, who spent a median 98.6% (IQR 92.8–100; n = 125) of their time *outdoors*. Estimates by reported reason for visit for each clinic are provided in Table H in S1 Text.

## Occupancy density of indoor spaces

Data from three clinics were sufficient to estimate occupancy density of at least two indoor spaces (Fig 2). In clinic KZN6 (Fig 2, panel 2), the occupancy density of area A (the main waiting area) consistently declined over the course of the day as individuals moved into areas B and C (smaller, 'downstream' waiting areas). Because of its relatively large volume, the occupancy density of area A never went above 0.9 persons/m$^2$. In contrast, in the smallest space (area C), occupancy peaked at around 1200, with density around or above 2.0 persons/m$^2$ from 1000–1200. In clinic KZN2 (Fig 2, panel 3), the smaller overall numbers of attendees meant that although the spaces are of similar size to clinic KZN6, density was generally lower. Overall occupancy was highest in clinic WC1, but the larger waiting spaces in this clinic meant that occupancy density was never higher than 1.0 persons/m$^2$ (Fig 2, panel 4), even in the smallest space. Clinic WC1 also had a functioning date-time appointment system, which is likely why occupancy of these spaces was more evenly distributed over the day compared with the other two clinics.

Occupancy density by room volume (persons/m$^3$) was calculated for the same spaces (Table I in S1 Text). This is a more relevant measure of occupancy density for pathogens transmitted predominantly via aerosol, such as *Mtb*. All assessed waiting spaces in clinics KZN2 and KZN6 had relatively low ceilings (maximum height 2.5–2.7 m) and occupancy density was higher (median 0.21–1.02 persons/m$^3$) than in spaces in clinic WC1, where ceilings were higher (maximum height 4.2–5.9 m; median occupancy density 0.10–0.14 persons/m$^3$).

## Discussion

We tracked 2,903 clinic attendees at 11 PHC clinics in two provinces of South Africa. Median time spent in clinic was 2 hours 36 minutes (IQR 01:36–03:43). People who arrived early in the morning spent longer in clinic, as did women and individuals attending with babies. Individuals attending for TB and maternal care spent less time in clinic. People attending clinics that had outdoor covered waiting areas spent more of their visit time outdoors, though differences were also seen between individuals attending the same clinic based on how care was organised for different 'streams'. In clinics with multiple indoor waiting areas, occupancy was often not distributed evenly between areas or over time; periods of high occupancy density (>2 persons/m$^2$) were observed in smaller waiting areas.

Time spent in clinic was below the national target maximum time [29] of three hours for around 60% of clinic attendees (ranging from 48% to 82% across clinics), but was over four hours for around 20% (range 7%–37%) and over five hours for around 9% (range 4%–27%). Detailed comparison with other studies is challenging given the variation in operational characteristics of PHC clinics and methods used (Table J in S1 Text). On crude comparison, median time spent in clinic in our study was slightly higher than seen in recent South African studies (Stime et al. [urban KZN, 2016] [24], median 01:48 for sexually transmitted infection

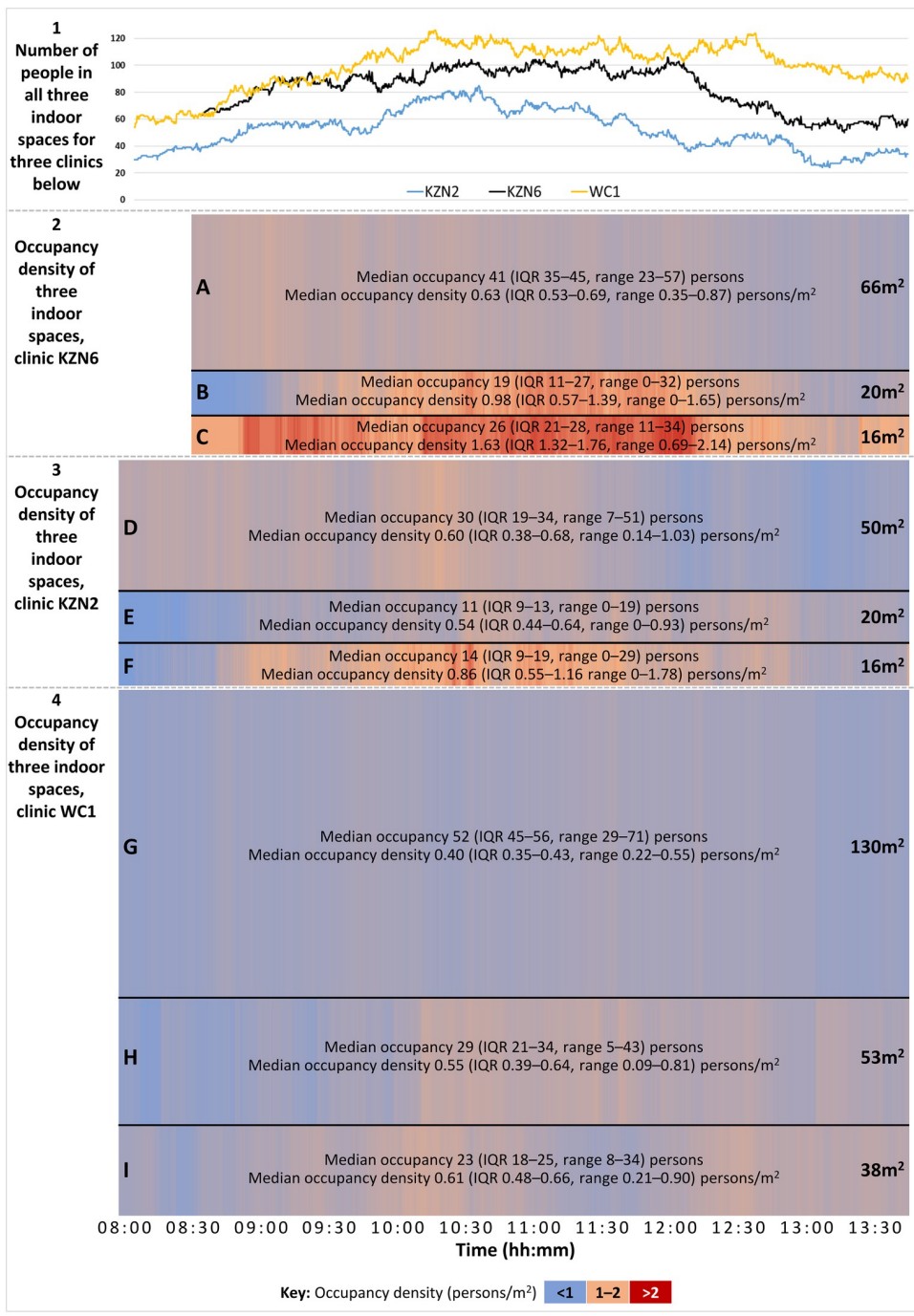

**Fig 2. Line graph (panel 1) and heat maps (panels 2–4) showing, respectively, total numbers of people in three indoor waiting areas and approximate occupancy density (in persons/m$^2$) of each waiting area in each of clinics KZN2, KZN6, and WC1 from 0800–1345*.** *Data available only from 0830 for clinic KZN6. Height of each row proportional to the area of the space. Each clinic was visited on a different day. See Table H in S1 Text for occupancy density relative to room volume (persons/m$^3$). Total numbers (line graph) indicative only of numbers of people occupying the three spaces examined, not overall numbers of people in the entire clinic. Spaces A, D, and G were the main (pre-filing +/- pre-vitals) formal waiting areas for their respective clinics; spaces B, C, H, and I were formal (pre-vitals and/or pre-consultation) waiting areas; space E was a corridor used as a pre-consultation waiting area; and space F was a combined pre-vitals waiting area, vitals administration area, and patient registration area. hh: hours; IQR: interquartile range; mm: minutes.

care and median 02:46 for HIV care; Egbujie et al. [rural KZN, 2014] [43], median 01:56 in nine PHC clinics) and slightly lower than seen in older studies (Bachmann and Barron [urban WC, 1997] [23], median 2.6 hours and 4.1 hours for 'preventive' and 'curative' care, respectively). Patterns in our data were also observed by previous investigators, including longer times for individuals who arrived earlier [23, 24, 43] and the early arrival of the majority of attendees, often before the clinic opened [30]. A higher patient to nurse ratio was strongly associated with longer waiting times in the study by Egbujie et al. [43], but not in our study, possibly because our estimates of staff numbers included all clinical staff, not only nurses. We are not aware of any previous studies that estimated proportions of time spent indoors vs. outdoors or the occupancy density of waiting areas.

Early arrival and queueing outside clinics is common in South Africa. It is influenced by the frequent absence of appointment and queue management systems; the organisation of services around the 'morning rush'; the lack of incentives for staff to change working patterns; and a variety of factors outside the health system, such as the availability of public transport and the community's trust in the health system. Detailed exploration of these issues is beyond the scope of this paper, but some discussion can be found in the report of an *Umoya omuhle* workshop on patient flow that involved a range of South African experts [44].

The observed between-clinic and within-clinic variation in proportions of time spent indoors vs. outdoors reflects the importance of both clinic design and the organisation of care in moderating the risk of respiratory disease transmission in these settings. The existence of an outdoor, 'low risk' waiting area is of little benefit if most individuals spend most of their time in poorly ventilated indoor spaces. In this study, individuals spent most of their time indoors, even in clinics with outdoor waiting areas, and this may be a missed opportunity to reduce transmission risk. However, thermal comfort and user preferences are important considerations when planning changes to patient flow, and the use of outdoor spaces may be less feasible in areas with lower temperatures (such as South Africa's Western Cape province).

In clinics where it may be impractical to wait outdoors, risk indoors can be moderated through more even distribution of occupancy throughout the available space. For example, during the busiest period in a small clinic like KZN6 (106 people in the clinic [Fig 2]), restricting occupancy of the smaller waiting spaces (B and C) to 20 and 16 individuals, respectively, would have left 70 individuals in the largest space and resulted in an occupancy density of around 1.0 persons/m$^2$ in all three spaces. This is in line with 2014 WHO guidelines for spatial separation as part of IPC for 'epidemic- and pandemic-prone acute respiratory infections', which recommend maintaining a distance of at least 1 metre between patients [15].

South African draft national guidelines [45] suggest a number of potential interventions to reduce waiting times and improve patient flow. Some have been tested in South Africa and other similar settings and are discussed briefly below.

## Potential interventions

Interventions to improve flow can be classified broadly as targeting two domains: 1) reducing the number of attendees overall and/or in particular spaces; and 2) reducing the time spent by attendees overall and/or in particular spaces. Most measures affect both domains, sometimes indirectly.

Initiatives to reduce numbers of attendees include the Central Chronic Medicine Dispensing and Distribution (CCMDD) system, where certain groups of patients collect chronic medication from community-based sites [17, 46], and reducing the frequency of routine clinic visits for certain conditions, for example by increasing the amount of medication provided (trials among people taking ART have shown promising results) [16, 47–50].

Measures to improve the overall efficiency of the clinic aim to move people through the facility as quickly as possible and to reduce the likelihood of bottlenecks in flow. These include holistic approaches, such as 'Lean' [51, 52], value-stream mapping [53], and other quality improvement methods [54], as well as more targeted changes in staffing or resources at specific points in clinical pathways [24].

Streaming and triage interventions focus on the movement of people once they enter a health facility. In line with Ideal Clinic guidance [28], every clinic in our study operated a streaming system that allowed people attending for TB care to bypass many of the steps in the pathway. This is partly intended to reduce the risk of *Mtb* transmission (although a high proportion of within-clinic transmission is likely to be driven by people with undiagnosed disease in general waiting areas) [55, 56] and is made feasible by the relatively small numbers of people treated for TB at each clinic and because no additional triage process is required. Triage (broadly defined as the process of prioritising patients for care based on their needs) [57] has also been shown to reduce waiting times in a hospital in South Africa, though it was less effective when used in two PHC clinics [58, 59]. Effective triage can be challenging and resource-intensive to sustain [60], and sub-optimal implementation of symptom-based triage for TB IPC has been documented by several studies [61–64]. Active queue management has also been tested: a qualitative study around the use of a 'Fast Queue' in clinics in KZN found that the use of multiple, managed queues was generally well-received by attendees, particularly if accompanied by smooth (i.e., unidirectional) flow and effective communication with health workers, though there were still those who experienced long waiting times [65].

Date-time appointment systems have been most widely used to reduce both numbers of people and time spent in clinics. Our study provides circumstantial evidence that appointment systems may help improve patient flow, and they have been shown to reduce waiting times in outpatient ART clinics in Ethiopia [66] and Kenya [67], antenatal clinics in Mozambique [68], and PHC clinics in South Africa [43], the last as part of a suite of interventions that included streaming, training, and infrastructure upgrades. Investigators describe generally encouraging results, though they also highlight the considerable challenges involved in standardising implementation at facilities that are differently organised. During the *Umoya omuhle* patient flow workshop, discussions around appointment system implementation emphasised the importance of support processes (such as pre-retrieval of files) and technological infrastructure in sustaining this complex intervention [44].

## Recommendations to improve patient flow

Building flexibility into the organisation of flow would allow a clinic to adapt to and absorb periods of increased traffic without putting patients or staff at risk; for example, by moving people from an overcrowded area to one that is relatively empty, or by activating 'overflow' covered outdoor waiting areas. However, any such initiative would require 1) a queue management system, to ensure that individuals moved between areas would not be placed at a disadvantage, and 2) clinic managers to have a) easy access to real-time information about flow and b) the resources and freedom to try to improve flow [27]. Patient flow can be difficult to measure quickly: previous published descriptions focus on largely qualitative descriptions of observed movement patterns [23, 69]. Occupancy density, however, is easy to measure (e.g., through manual headcounts) and, measured periodically across a clinic, could be used as a proxy estimate for flow. We suggest that regular, light-touch ('diagnostic') approximation of this metric may have numerous potential direct and indirect benefits, including improved efficiency; shorter waiting times; better clinic-specific decision-making; and a strengthened relationship between the clinic and its community [27, 44, 70].

Importantly, interventions intended to reduce attendance and waiting times may adversely affect the flow around which the clinic was designed and may therefore increase the rate of transmission to an individual during the time they do spend in the clinic. Most clinics are designed with waiting areas that get successively smaller as patients move through the pathway; as pathways diverge, patients 'diffuse' through the clinic and one would expect occupancy to be lower. However, if the overall 'patient load' is greater than the capacity of the clinic, or if different stages of the pathway are variably efficient, or if certain attendees (e.g., those with appointments) are allowed to skip parts of the queue, bottlenecks can arise in areas that are designed to hold fewer people, leading to higher than optimal occupancy of 'downstream' areas and/or under-use of 'upstream' areas. Interventions to improve flow and reduce waiting times are acutely vulnerable to achieving "many small successes and one big failure" [71] and should be undertaken with careful consideration of potential effects on other parts of the pathway, possible increases in risk of disease transmission, and adjustments that may be needed in resource allocation, ventilation, and the organisation of care.

## Strengths and limitations

The method employed in this study was relatively inexpensive, built on an approach already widely used in South African PHC clinics, and included elements that could be incorporated into routine estimation of waiting times and flow. Numbers of individuals who declined to participate were not recorded and we were therefore unable to assess for selection bias introduced by the enrolment process. Starting data collection after some individuals had arrived and stopping data collection at 1400 (because of logistical restrictions) reduced the numbers of individuals whose data could be used to estimate total waiting time, requiring the use of multiple imputation to deal with missing data and reduce the risk of bias. Multiple imputation assumes that the data are missing at random, which means that the observed values can be used to predict the missing values. However, if the assumption is incorrect, the results may be biased. Furthermore, the validity of results derived from multiply imputed data depend on the appropriateness of the imputation model. Future similar exercises should, at minimum, continue to record clinic exits for as long as possible. Although we attempted to collect information on whether attendees had appointments, the data were of very low quality and we were unable to include this variable in the regression analysis. Because of 1) the amount of missing data, 2) the variability between and within clinics, and 3) because data were collected on only one day from almost all clinics (seasonal changes in symptom/disease prevalence are also likely to affect rates of clinic attendance and therefore patient flow and waiting times), reported estimates should not be considered representative of the two provinces, types of clinics, or the clinics themselves. In busy clinics in particular, many attendees' barcodes were not scanned every time at every scanning point, and estimates of waiting area occupancy and time spent indoors or outdoors should be treated as approximations. Even so, our headline findings are plausible and consistent with those from other studies.

## Conclusions

Measuring patient flow is important for estimating clinic efficiency and risk of disease transmission. In our study, females, individuals arriving early, and those attending with young children spent longer at clinic. Attendees generally waited where they were asked to: using outdoor waiting areas as part of designated patient pathways increased the proportion of visit time spent outdoors, though most individuals still spent most of their time indoors, suggesting that outdoor spaces could be better utilised. Occupancy of indoor spaces varied considerably over the day and people often were not distributed evenly throughout the available space.

Regular, light-touch estimation of occupancy density may help staff to quickly assess patient flow and guide the use of interventions to improve it.

## Supporting information

**S1 Text.**
(PDF)

## Acknowledgments

Our thanks to the managers, staff, and attendees at all study clinics for their participation, enthusiasm, and patience; to Dr Bart Willems for guidance around methods; to Dr Gavin Reagon for additional advice, particularly around appointment systems; and to all participants of the *Umoya omuhle* participatory workshop on waiting times and patient flow [44], held in South Africa in August 2019.

Special thanks to teams in Somkhele and Cape Town who were involved in data collection: Amy Burdzik, Anathi Mngxekeza, Awethu Gawulekapa, Duduzile Mkhwanazi, Emmerencia Gumede, Godfrey Manuel, Nompilo Ndlela, Nonhlanhla Madlopha, Nozipho Mthethwa, Phumzile Nywagi, Precious Mathenjwa, Samantha Moyo, Seonaid Kabiah, Sinead Murphy, Siphokazi Adonisi, Siphosethu Titise, Sithembiso Luthuli, Sphiwe Mthethwa, Suzanne Key, Tamia Jansen, Thandekile Nene, Yolanda Qeja, Yutu Dlamini, and Zinhle Mkhwanazi.

*Umoya omuhle* was a multidisciplinary initiative involving several institutions and a team of over 100 people (Table K in S1 Text).

## Author Contributions

**Conceptualization:** Aaron S. Karat, Nicky McCreesh, Karina Kielmann, Hayley MacGregor, Anna Vassall, Tom A. Yates, Alison D. Grant.

**Data curation:** Aaron S. Karat, Kathy Baisley.

**Formal analysis:** Aaron S. Karat, Nicky McCreesh, Kathy Baisley.

**Funding acquisition:** Karina Kielmann, Hayley MacGregor, Anna Vassall, Tom A. Yates, Alison D. Grant.

**Investigation:** Aaron S. Karat, Nicky McCreesh, Indira Govender, Idriss I. Kallon, Hayley MacGregor, Anna Vassall, Tom A. Yates, Alison D. Grant.

**Methodology:** Aaron S. Karat, Nicky McCreesh, Kathy Baisley, Indira Govender, Anna Vassall, Alison D. Grant.

**Project administration:** Karina Kielmann, Alison D. Grant.

**Resources:** Aaron S. Karat, Anna Vassall, Tom A. Yates, Alison D. Grant.

**Supervision:** Kathy Baisley, Karina Kielmann, Hayley MacGregor, Anna Vassall, Alison D. Grant.

**Visualization:** Aaron S. Karat.

**Writing – original draft:** Aaron S. Karat.

**Writing – review & editing:** Aaron S. Karat, Nicky McCreesh, Kathy Baisley, Indira Govender, Idriss I. Kallon, Karina Kielmann, Hayley MacGregor, Anna Vassall, Tom A. Yates, Alison D. Grant.

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
