## [Decision Letter · Decision Letter 0]

8 Feb 2022

PGPH-D-21-00590

Waiting times, patient flow, and occupancy density in South African primary health care clinics: implications for infection prevention and control

Dear Dr. Karat

Thank you for submitting your manuscript to PLOS Global Public Health. After careful consideration, we feel that it has merit but does not fully meet PLOS Global Public Health’s publication criteria as it currently stands. Therefore, we invite you to submit a revised version of the manuscript that addresses the points raised during the review process.

Please revise the discussion and the conclusion to reflect the data in the results section. Otherwise revise the title of the article. The data currently presented does not support the study title and the discussion and conclusions are not aligned to the study title/topic. 

Please submit your revised manuscript by 21/February/2021.  If you will need more time than this to complete your revisions, please reply to this message or contact the journal office at globalpubhealth@plos.org. Please include the following items when submitting your revised manuscript:

We look forward to receiving your revised manuscript.

Kind regards,

Reuben Kiggundu

Academic Editor

Journal Requirements:

1. In your ethics statement in the manuscript , please state whether the IRB or ethics committee waived the requirement for informed consent.

3. Please amend your detailed Financial Disclosure statement. This is published with the article, therefore should be completed in full sentences and contain the exact wording you wish to be published.

State what role the funders took in the study. If the funders had no role in your study, please state: “The funders had no role in study design, data collection and analysis, decision to publish, or preparation of the manuscript.”

4. Please update your Competing Interests statement. If you have no competing interests to declare, please state: “The authors have declared that no competing interests exist.”

5. In the online submission form, you indicated that your data will be submitted to a repository upon acceptance.  We strongly recommend all authors deposit their data before acceptance, as the process can be lengthy and hold up publication timelines. Please note that, though access restrictions are acceptable now, your entire data will need to be made freely accessible if your manuscript is accepted for publication. This policy applies to all data except where public deposition would breach compliance with the protocol approved by your research ethics board. If you are unable to adhere to our open data policy, please kindly revise your statement to explain your reasoning and we will seek the editor's input on an exemption. Please be assured that, once you have provided your new statement, the assessment of your exemption will not hold up the peer review process.

6. We have noticed that you have uploaded supporting information but you have not included a list of legends. Please add a full list of legends for all supporting information files (including figures, table and data files) after the references list.

Additional Editor Comments:

The link between the study data (patient waiting time, etc.) is not clearly linked to the IPC.

The data collection and methods sections do not describe if any data about IPC outcomes/indicators was collected. How then do the authors link their findings to IPC? It seems IPC was not part of the study and the authors are making inferences in regards to IPC based on the findings of patient flow in the clinic. If this is the case, I suggest we add a statement in the discussion sections so we have this clarity.

Revise the recommendation section to show the connection between the study topic and IPC.

Currently, the title of the paper is not reflected in the discussion section and the recommendations. If this revision is not possible, please consider revising the title of the manuscript.

Additionally, if we maintain IPC in the title, the authors should add the word TB, so that it can read as ‘’TB IPC’’.

The authors should discuss the implications of their findings to IPC. For example, they state that ‘’early arrival, being female, and attending with a young child lead to longer wait times? However, it would be important to discuss the implications of these variables on the IPC.

Reviewers' comments:

Reviewer's Responses to Questions

**Comments to the Author**

1. Does this manuscript meet PLOS Global Public Health’s publication criteria? Is the manuscript technically sound, and do the data support the conclusions? The manuscript must describe methodologically and ethically rigorous research with conclusions that are appropriately drawn based on the data presented.

Reviewer #1: Partly

Reviewer #2: Partly

2. Has the statistical analysis been performed appropriately and rigorously?

Reviewer #1: Yes

Reviewer #2: I don't know

3. Have the authors made all data underlying the findings in their manuscript fully available (please refer to the Data Availability Statement at the start of the manuscript PDF file)?

Reviewer #1: Yes

Reviewer #2: Yes

4. Is the manuscript presented in an intelligible fashion and written in standard English?

Reviewer #1: Yes

Reviewer #2: Yes

5. Review Comments to the Author

Reviewer #1: The manuscript is technically sound, although there are a few areas where further clarification of limitations of the data and the analytical approach could be used to support the conclusions draw (and their relative strength). Further details have been provided in the attachment.

Analyses conducted by the authors have been performed appropriate and rigorously, and all data underlying will be made available through LSHTM DataCompass.

The manuscript is well organized and well written

Please find additional comments for the author in the attachment.

Reviewer #2: Conceptually, it is understandable why reducing overcrowding in healthcare facilities is important to IPC measures, unfortunately, the authors fail to make an argument for why this specific study is necessary, novel, or innovative. It is not clear to me that this study was actually needed to draw the conclusions and recommendations described in this paper. Regarding specific conclusions: (1) the finding that patients who were attending clinic for TB care spent less time in clinic is not novel if there is a built-in fast track system and (2) the finding that patients designated to outdoor spaces spent more of their visit time outdoor is not novel. Again, it is not clearly stated why this study was necessary to draw these specific types of conclusions.

Additionally, I am concerned by the significant amount of missing data. While, I understand that multiple imputation was used to generate arrival and/or departure times for individuals for whom the data was not recorded, why did the group not expand their daily data collection time period so that they were present to ensure that the majority of patient arrival/departure times were recorded?

Additional Questions:

Why did the team choose occupancy density as a proxy for patient flow? Is this a standard proxy in the field and if not, why was this chosen for this study’s outcome?

How many days of data collection were done at each site? Was it daily for each site in KZN and WC? This is not clear in the data collection section.

Data was collected at different times of the year for each region. How does the time period when data was collected affect findings at each site—i.e., seasonal volume, primary chief complaints by season, weather (particularly for clinics with outdoor waiting spaces)?

Why was FIVE minutes selected for the total capture time cut off for exclusion? Are there folks that really make it in and out of clinic in less than say 10 minutes?

Why does it matter that early arrival, being female, and attending with a young child lead to longer wait times? Are there significant clinical outcomes to these findings? Are there practical changes that need to be made in patient flow for these groups? This is not directly addressed.

If appointment systems have been shown to reduce waiting times and the study included clinics with appointment system, I would have liked to see some discussion of how the study’s primary outcomes compared among clinics with appointment systems and those without. What percentage of the WC patients were part of the date-time appointment system? Was this a significant amount?

“In four clinics with an outdoor waiting area that was used as part of a designated patient pathway, individuals spent a median 13.7% (IQR 242 1.4–74.5; n = 1,362) of their time outdoors, compared with a median 0% (IQR 0–1.4; n = 828) outdoors among attendees at the five clinics that did not have an outdoor waiting area or where the outdoor area was not used.” Why did such a small percentage of patients with a designated outdoor waiting area not wait outdoors? If patients do not spend much time in the waiting area overall, does it really matter if more clinics make more outdoor waiting space available?

6. PLOS authors have the option to publish the peer review history of their article (what does this mean?). If published, this will include your full peer review and any attached files.

**Do you want your identity to be public for this peer review?** For information about this choice, including consent withdrawal, please see our Privacy Policy.

Reviewer #1: No

Reviewer #2: No

---

## [Editor Report · Decision Letter 1]

13 Jun 2022

Estimating waiting times, patient flow, and waiting room occupancy density as part of tuberculosis infection prevention and control research in South African primary health care clinics

PGPH-D-21-00590R1

Dear Aaron

We are pleased to inform you that your manuscript 'Estimating waiting times, patient flow, and waiting room occupancy density as part of tuberculosis infection prevention and control research in South African primary health care clinics' has been provisionally accepted for publication in PLOS Global Public Health.

Best regards,

Reuben Kiggundu

Academic Editor
